# Strong Early Impact of Letrozole on Ovulation Induction Outperforms Clomiphene Citrate in Polycystic Ovary Syndrome

**DOI:** 10.3390/ph17070971

**Published:** 2024-07-22

**Authors:** Rita Zsuzsanna Vajna, András Mihály Géczi, Fanni Adél Meznerics, Nándor Ács, Péter Hegyi, Emma Zoé Feig, Péter Fehérvári, Szilvia Kiss-Dala, Szabolcs Várbíró, Judit Réka Hetthessy, Levente Sára

**Affiliations:** 1Department of Obstetrics and Gynecology, Semmelweis University, 1088 Budapest, Hungary; gandrasmihaly@gmail.com (A.M.G.); acs.nandor@med.semmelweis-univ.hu (N.Á.); varbiroszabolcs@gmail.com (S.V.); saralevente@icloud.com (L.S.); 2Centre for Translational Medicine, Semmelweis University, 1085 Budapest, Hungary; f.meznerics@gmail.com (F.A.M.); hegyi2009@gmail.com (P.H.); feigemma@gmail.com (E.Z.F.); fehervari.peter.biomat@gmail.com (P.F.); kissdalaszilvia.tm@gmail.com (S.K.-D.); 3Department of Dermatology, Venereology and Dermatooncology, Semmelweis University, 1085 Budapest, Hungary; 4Institute of Pancreatic Diseases, Semmelweis University, 1083 Budapest, Hungary; 5Institute for Translational Medicine, University of Pécs, 7624 Pécs, Hungary; 6Department of Biomathematics and Informatics, University of Veterinary Medicine, 1400 Budapest, Hungary; 7Department of Orthopaedics, Semmelweis University, 1082 Budapest, Hungary; drhjr612@gmail.com

**Keywords:** clomiphene citrate, letrozole, ovulation induction, polycystic ovary syndrome, subendometrial blood flow

## Abstract

Polycystic ovary syndrome is a common endocrine disorder, characterized by hyperandrogenism and/or chronic oligo/anovulation, which leads to infertility. The aim of this systematic review and meta-analysis was to explore the efficacy of letrozole compared with clomiphene citrate for ovulation induction in women with polycystic ovarian syndrome. The study protocol has been registered with PROSPERO (registration number CRD42022376611). The literature search included randomized clinical trials. We conducted our systematic literature search across three medical databases: MEDLINE (via PubMed), Cochrane Library (CENTRAL), and Embase. The data synthesis employed a random effects model. Out of the 1994 articles screened, 25 studies fulfilled the inclusion criteria. The letrozole group exhibited a significant increase in endometrial thickness (mean difference = 1.70, confidence interval: 0.55–2.86; I^2^ = 97%, *p*-value = 0.008). The odds of ovulation (odds ratio = 1.8, confidence interval: 1.21–2.69; I^2^ = 51%, *p*-value = 0.010) and pregnancy (odds ratio = 1.96, confidence interval: 1.37–2.81; I^2^ = 32%, *p*-value = 0.002) were significantly higher. The resistance index of the subendometrial arteries showed a significant decrease (mean difference = −0.15, confidence interval: −0.27 to −0.04; I^2^ = 92%, *p*-value = 0.030). Women diagnosed with polycystic ovarian syndrome and treated with letrozole for ovulation induction had increased ovulation and pregnancy rates and increased endometrial thickness. The lower resistance index of subendometrial arteries can enhance intrauterine circulation, creating more favorable conditions for embryo implantation and development.

## 1. Introduction

Polycystic ovary syndrome (PCOS) is among the most prevalent endocrinological issues contributing to female infertility globally [1,2,3,4]; the prevalence of this condition ranges from 9% to 18% among fertile women [5]. The primary features of the disease include hyperandrogenism and/or chronic oligo/anovulation, leading to infertility [6,7].

Clomiphene citrate (CC) has been employed as a first-line medication for ovulation induction (OI) in women with PCOS since 1960 [8,9]. However, it is associated with some undesirable and non-dose-dependent side effects, including hot flashes, an increase in ovarian size, bloating, nausea and vomiting, breast sensitivity and pain, headaches, hair loss, insomnia, and depression [10]. Due to the longer half-life of CC, pregnancy may not occur despite ovulation, possibly attributed to its antiestrogenic effects on the endocervix and the endometrium [5,8,11].

Letrozole (LE) is a third-generation aromatase inhibitor, approved as an adjuvant in the treatment of estrogen receptor-positive breast cancer [12,13,14,15]. LE does not exhibit anti-estrogenic effects [16,17] and is associated with minor side effects such as leg cramps and headaches [18,19]. Compared to CC, LE has a shorter half-life and is cleared from the circulation more rapidly [8]. Aromatase inhibition with LE leads to up-regulation of endometrial receptors and rapid endometrial growth without adverse effects on endometrial receptivity [8,15]. Improved blood supply to the subendometrial halo and the endometrium results in a thicker endometrial wall, thereby maximizing the chances of pregnancy [20].

Despite the numerous beneficial properties of LE, it has not yet been included in the first-line therapy for PCOS patients. Limited information is available regarding its effectiveness. Therefore, we aimed to assess the efficacy of LE in improving ovulation rate, endometrial receptivity, and pregnancy rate in women with PCOS compared to the effects of CC through a systematic review and meta-analysis.

Compared to previously published studies, a fundamental innovation in our article is that we specifically examined the early effect of letrozole versus clomiphene citrate. To get a more accurate picture of the mechanism of action of the two active substances, we only compared studies in which only clomiphene citrate or letrozole was administered to patients with PCOS and in which the drugs mentioned above were used in a fixed dose during one cycle. Previous studies published on a similar topic showed a significant deviation in terms of drug dose and therapy, which does not allow for an exact comparison of these active substances and distorts the quantitative analysis.

## 2. Results

### 2.1. Search and Selection

We identified 1994 articles through our systematic search. After removing duplicate articles, we reviewed the titles and abstracts of 1274 articles. Ultimately, we included 22 studies [1,4,15,16,19,21,22,23,24,25,26,27,28,29,30,31,32,33,34,35,36] after the full-text selection, as well as three additional studies found through a citation search [5,6,37]. The details of the search and selection process are illustrated in Figure 1.

### 2.2. Characteristics of the Included Studies

The baseline and patient characteristics of the studies included in the systematic review and meta-analysis are outlined in Table 1A,B.

### 2.3. The Results of the Quantitative Analysis

Based on the results of 11 studies involving 1651 patients, endometrial thickness (ET) was significantly higher in the LE group/mean difference (MD) = 1.70, confidence interval (CI): 0.55–2.86; I^2^ = 97%, *p*-value = 0.008/compared to the CC group (Figure 2A). The subgroup analysis conducted on ovulating patients showed a thicker endometrium in the LE group; however, this difference did not reach statistical significance (MD = 2.2, CI: −0.38–4.78; I^2^ = 97%, *p* = 0.077) (Appendix A). 

Based on the results of 9 studies including 1510 patients, the odds of ovulation, and based on the results of 11 studies involving 1410 patients, the odds of pregnancy were significantly higher among patients receiving LE compared to those receiving CC (ovulation rate: odds ratio (OR) = 1.80, CI: 1.21–2.69; I^2^ = 51%, *p* = 0.010 (Figure 2B); pregnancy rate: OR = 1.96, CI: 1.37–2.81; I^2^ = 32%, *p* = 0.002 (Figure 2C)). In patients who ovulated as a result of the drug therapy, the odds of pregnancy were also higher in the LE group, although this difference did not reach statistical significance (OR = 1.65, CI: 0.40–6.76; I^2^ = 56%, *p* = 0.337) (Appendix A).

Based on the results of three studies involving 460 patients, the resistance index (RI) of subendometrial arteries was significantly lower in patients treated with LE compared to those treated with CC (MD = −0.15, CI: −0.27–−0.04; I^2^ = 92%, *p* = 0.030), as illustrated in Figure 3A. Additionally, the pulsatility index (PI) of subendometrial arteries was also lower in the LE group (MD = −0.17, CI: −0.81–0.47; I^2^ = 95%, *p* = 0.372), although this disparity did not attain statistical significance (Figure 3B).

Based on the results of eight studies including 253 patients, the odds of multiple pregnancies were higher in patients treated with CC compared to those treated with LE (multiple pregnancies: OR = 0.41, CI: 0.12–1.35; I^2^ = 0%, *p* = 0.119), as illustrated in Appendix A. However, this disparity did not achieve statistical significance.

Based on the results of four studies including 160 patients, no significant difference was observed in the miscarriage rate (OR = 0.62, CI: 0.19–1.98; I2 = 0%, *p* = 0.278) (Appendix A).

No significant difference was observed between groups in the number of follicles (MD = −0.40, CI: −0.84–0.03; I^2^ = 91%, *p* = 0.066) based on the results of nine studies involving 1264 patients (Appendix A) and in the diameter of follicles (MD = 0.58, CI: −0.17–1.32; I^2^ = 45%, *p* = 0.092) based on the results of four studies involving 660 patients (Appendix A) despite a higher count of smaller follicles observed in the CC group. In the subset analysis focusing on patients who ovulated, no significant difference was observed between the two groups in terms of the number of follicles (MD = −0.80, CI: −2.48–0.89; I^2^= 80%, *p* = 0.179) (Appendix A).

Based on the results of three studies involving 563 patients, the **odds** were higher **for monofollicular development** and lower **for multifollicular development** in patients treated with LE compared to those treated with CC (monofollicular development: OR = 1.99, CI: 0.62–6.34; I^2^ = 51%, *p* = 0.126 (Appendix A); multifollicular development: OR = 0.50, CI: 0.16–1.61; I^2^ = 51%, *p* = 0.126 (Appendix A)), although this difference did not reach statistical significance.

### 2.4. The Results of the Qualitative Analysis

Table 2A,B outline the results and main conclusions of the studies exclusively included in the systematic review.

## 3. Discussion

PCOS and infertility resulting from PCOS represent significant global challenges, yet effective treatment remains elusive. Extensive research endeavors are ongoing to identify optimal drugs for ovulation induction; nevertheless, a comprehensive synthesis of study findings has been lacking. In our investigation, we consolidated results from existing publications and delineated several advantageous characteristics of LE as an ovulation induction agent compared to the current use of CC.

In our meta-analysis, ET was significantly greater in the LE group compared to the CC group. When juxtaposing our findings with previous meta-analyses, we saw that He et al. noted significant heterogeneity precluding data pooling [39]. However, Gadalla et al. reported a significantly thinner endometrium in the CC group [11]. Similarly, Akinoso-Imran et al. reported analogous results following LE treatment [40]. Among the RCTs scrutinized in our qualitative analysis (Table 2A,B), six revealed a significantly thicker endometrium in the LE group [15,23,32,33,34,35,38]. In comparison, three identified a significantly thicker endometrium in the CC group [28,29,31], and two detected no significant difference between the two groups [30,36]. Additionally, other studies noted significantly higher endometrial volume [1,5], a more frequent multilayered endometrial pattern [1,23], and the occurrence of hypoechogenic endometrium [23] in the LE group. Conversely, one study found no significant difference between the two groups concerning endometrial pattern and echogenicity [29].

Patients receiving CC appear to develop more—but smaller—follicles, potentially diminishing the likelihood of appropriate maturation and follicle rupture, and increasing the risk of hyperstimulation and multiple pregnancies [41]. We observed a higher number of dominant follicles in patients receiving CC, although the definition of dominant follicle varied among the studies (ranging from >12 mm to ≤18 mm). The diameters of follicles did not exhibit significant differences; however, follicles in the LE group tended to be more substantial.

Comparison with previous meta-analyses revealed that He et al. noted significantly fewer mature follicles per cycle in patients treated with letrozole [39]. Other studies found no statistically significant difference in the standardized mean difference between patients treated with LE or CC, but fewer mature follicles were found in the LE group [40]. Among the RCTs in our study, contradictory results were obtained (Table 2A,B) [28,29,30,32,33,35,38].

Ovulation is imperative for fertilization. We previously demonstrated that patients in the LE group exhibited significantly higher odds of ovulation than those in the CC group. In comparison to our findings, He et al. [39], Tsiami et al. [42], and Zhuo et al. [43] reported similar results. Wang et al. noted a significantly superior ovulation rate among patients receiving LE compared to CC [44]. Conversely, Roque et al. [45] and Gadalla et al. [11] found similar odds of ovulation in the LE and CC groups. Akinoso-Imran et al. also observed a significantly higher ovulation rate with letrozole treatment than with CC [40]. Similar results were reported by Shifu Hu et al. [46]. Among the RCTs analyzed in our qualitative assessment (Table 2A,B), three reported significantly higher ovulation rates in the LE group [31,33,34], while two found no significant difference between the two groups [32,36]. Another study found significantly higher rates of monofollicular development in the LE group [36].

It would be beneficial if more women with PCOS increased their chances of achieving pregnancy. We discovered that the odds of pregnancy were significantly higher in patients treated with LE than in the CC group. Among previous meta-analyses, four studies reported significantly higher pregnancy rates [40,43,44,45], and three studies noted considerably higher odds of live birth [43,44,45] after LE treatment. Gadalla et al. [11] and Hu et al. [46] also documented similar differences between the two groups. In contrast, He et al. found no significant difference in the odds of pregnancy between the two groups [39]. Among the RCTs analyzed in our qualitative assessment (Table 2A,B), four reported significantly higher pregnancy rates in the LE group [29,31,33,36]; the others found no significant difference between the two groups [28,30,32,34,35,38]. Three studies found significantly higher live birth rates in the LE group [31,32,33], and three studies found no significant difference between the two groups [29,30,37].

Multiple pregnancies increase the likelihood of potential complications, such as preterm birth, fetal growth restriction, etc. [47]. In our study, the odds of single pregnancies were higher among LE patients, while more multiple pregnancies occurred among CC patients; however, this difference did not reach statistical significance. Four previous meta-analyses reached a similar non-significant conclusion [39,40,44,46].

The objective should not only be to attain pregnancy but also to carry as many pregnancies to term as possible by women with PCOS. In our study, the odds of miscarriage were higher in the CC group, although this difference did not achieve statistical significance. Our findings align with previous meta-analyses in the literature [39,40,44,45,46]. Among the RCTs included in our qualitative analysis (Table 2A,B), three studies found no significant difference in the number of miscarriages between the two groups [31,32,33], In contrast, other studies reported either no difference or one or two cases in each group [28,30,35]. Regarding the prevalence of ectopic pregnancy, one study found no significant difference between the two groups [38]. Fetal anomalies were not observed in either group in two RCTs [29,35]. Legro et al. reported four anomalies in the LE group and one in the CC group [31]. Another study found one anomaly in the LE group and one in the CC group [33], while yet another reported no cases in the LE group and one in the CC group [37].

The RI of subendometrial arteries was significantly lower in the LE group compared to the CC group. The PI of subendometrial arteries was also lower in patients treated with LE, although this result was not statistically significant. Lower RI and PI values may enhance intrauterine blood supply, potentially leading to improved intrauterine embryonic development. Wang et al. conducted an RCT and found no difference in PI values between the LE and CC groups after Diane-35 and metformin pretreatment; hence, we excluded these articles from our meta-analysis. However, the RI of subendometrial arteries was significantly lower in the LE group [48]. Another RCT investigated the PI and RI of spiral arteries over six months; RI and PI were significantly lower in the letrozole group [49]. Among the RCTs in our qualitative analysis (Table 2A,B), one study also reported significantly lower RI and PI of subendometrial arteries in the LE group. However, they utilized an escalating drug dosage and examined multiple cycles. Thus, they were not included in our meta-analysis [38]. Two studies found significantly higher endometrial VI, FI, and VFI in the LE group, with no significant difference in RI and PI of uterine arteries between the two groups [1,5]. Another study noted a significantly lower detection rate of endometrial/subendometrial blood flow in the LE group [19]. Furthermore, one study reported significantly higher SV/DV of subendometrial arteries and a significantly negative correlation between VEGF concentration and endometrial RI in the LE group. In contrast, no such correlation was observed in the CC group [15]. Additionally, one study found significantly higher VEGF and integrin alpha vß3 concentrations in uterine fluid in the LE group [5].

We obtained results similar to most previous meta-analyses, with the distinction that we also emphasized a superior effect achievable with the use of LE compared to CC, even within one cycle. We are the first to employ a meta-analysis to validate that using LE might confer benefits for subendometrial circulation compared to CC, potentially elucidating the significant increase in pregnancy rates among women with PCOS treated with LE. The alterations mentioned above in subendometrial circulation represent early changes after just one cycle, potentially accounting for the differences in significance between RI and PI values. With prolonged usage, these differences also become significant for PI values.

## 4. Materials and Methods

We present our systematic review and meta-analysis following the recommendations outlined in the PRISMA (Preferred Reporting Items for Systematic Reviews and Meta-Analyses) 2020 guidelines (Appendix A) [50]. Additionally, we adhered to the recommendations provided in the Cochrane Handbook for Systematic Reviews of Interventions Version 6.1.0 [51]. The study protocol was registered on PROSPERO (registration number CRD42022376611; see https://www.crd.york.ac.uk/PROSPERO; accessed on 17 November 2022), and strict adherence to the protocol was maintained. Furthermore, we conducted a post hoc analysis, including 12 additional outcomes indicated in the population-intervention-control-outcome (PICO) framework, to offer a comprehensive overview of all potentially significant outcomes. For our quantitative analysis, we only included studies that used fixed drug doses over a single cycle to present the early apparent effects of letrozole and clomiphene citrate.

### 4.1. Literature Search and Eligibility Criteria

We conducted our systematic literature search across three medical databases: MEDLINE (via PubMed), Cochrane Library (CENTRAL), and Embase, from inception to 21 November 2022. The search query ‘(polycystic ovarian syndrome OR PCOS OR Stein-Leventhal syndrome) AND (letrozole OR aromatase inhibitor)’ was applied to all fields in the search engines. No language or other restrictions were imposed. RCTs comparing LE to CC were included for patients diagnosed with PCOS based on the Rotterdam criteria (at least two years after menarche) [23]. The following PICO framework was applied:P: women with PCOSI: LEC: CCO: outcomes according to the protocol: endometrial thickness (ET); number of dominant follicles; ovulation rate; pregnancy rate; endometrial volume; additional outcomes: endometrial pattern- and echogenicity; diameter of dominant follicles; rate of mono- and multifollicular development; single and multiple pregnancy rate; live birth rate; miscarriage rate; prevalence of ectopic pregnancies; number of fetal anomalies; endometrial vascularization index (VI), flow index (FI), vascularization flow index (VFI), and detection rate of endometrial–subendometrial blood flow; resistance index (RI) and pulsatility index (PI) of subendometrial and uterine arteries; systolic velocity (SV)/diastolic velocity (DV) of subendometrial arteries; biomarker/vascular endothelial growth factor (VEGF) and integrin alpha vß3/concentrations in uterine fluid.

Reviews, case series, case reports, and studies including patients without a PCOS diagnosis, girls before menarche, and those within two years post-menarche were excluded from the study.

### 4.2. Study Selection and Data Collection

Two reviewers (RZSV and EZF) conducted the selection process independently. Data from the eligible articles were independently collected by two authors (RZSV and AMG). The following information was extracted regarding the study: first author, study type, year of publication, study population, and study period. Additionally, the following data were extracted concerning the patients studied: age, body mass index, duration of infertility, use of human chorionic gonadotropin for ovulation support or progesterone for luteal phase support, the timing of endometrial testing in relation to the cycle, dosage and timing of administration of LE or CC in relation to the cycle, and outcome data.

### 4.3. Study Risk of Bias Assessment

Two authors (RZSV and AMG) independently conducted the risk of bias assessment. The risk of bias was evaluated using the Risk of Bias Tool 2 (ROB2) [52], following the recommendations of the Cochrane Collaboration. The ROB2 tool is an instrument used to assess the risk of bias in the results of RCTs. It evaluates five domains: domain 1 assesses bias arising from the randomization process, domain 2 assesses bias due to deviations from intended interventions, domain 3 assesses bias due to missing outcome data, domain 4 assesses bias in the measurement of the outcome, and domain 5 assesses bias in the selection of the reported result.

The quality assessment of the included studies was performed according to the “Grades of Recommendation, Assessment, Development, and Evaluation (GRADE)” workgroup with GRADE-Pro, as recommended by the Cochrane Collaboration [53].

### 4.4. Synthesis Methods

Statistical analysis was conducted using R version 4.1.2 [54] along with the meta [55] and dmetar packages 6.1.0 [55].

Comparison between LE and CC involved calculating the mean differences for ET, the number and diameter of dominant follicles, and RI and PI values of subendometrial artery between the intervention and control groups. Additionally, ovulation rate, pregnancy rate, single pregnancy rate, the frequency of miscarriages, and the frequency of monofollicular and multifollicular development were examined. Furthermore, the following outcomes were separately analyzed in patients who ovulated only: ET, number of dominant follicles, and pregnancy rate.

For binary outcomes, odds ratios with 95% confidence intervals (CIs) served as outcome measures. The number of patients and events was either extracted or calculated, and the results are presented as the odds of an event in the LE group compared to the odds in the CC group.

Regarding continuous data, mean differences with 95% CIs were utilized. To compute mean differences (MDs), sample sizes, and mean and standard deviation values were extracted from the studies. MDs were calculated by subtracting the mean values of the CC group from those of the LE group.

The pooled odds ratio (OR) was calculated using the Mantel–Haenszel method [56] and the Robins, Greenland, and Breslow approach [57]. For zero cell counts, the exact Mantel–Haenszel method [56] (without continuity correction) was employed, following recommendations by Cooper, Hedges, and Valentine [58] as well as J. Sweeting, J. Sutton, and C. Lambert [59]. Confidence intervals were constructed using the Paule–Mandel method [60], as suggested by Veroniki et al. [61]. In instances of zero cell counts, individual study odds ratios with 95% CIs were computed by adding 0.5 as a continuity correction (this correction was solely utilized for visualization on forest plots).

Pooled MDs were computed using the inverse variance method.

CIs were calculated using the restricted maximum likelihood estimator with the Q profile method, following recommendations by Harrer et al. [62] and Veroniki et al. [61]. Additionally, Hartung–Knapp adjustments were applied [63] as suggested by IntHout, Ioannidis, and Borm [64].

Between-study heterogeneity was assessed using Higgins and Thompson’s I^2^ statistic [65] and the Cochrane Q test, as recommended by Harrer et al. [62]. The I^2^ statistic indicates the percentage of heterogeneity that cannot be attributed to random chance. Heterogeneity is considered substantial if I^2^ exceeds 75%.

### 4.5. Publication Bias and Heterogeneity

Publication bias was assessed using funnel plots and Egger’s test [66,67]. Specifically, for ET and pregnancy rate, funnel plots did not suggest publication bias, and Egger’s test results were not significant, indicating an absence of detectable publication bias.

However, substantial heterogeneity was observed for ET, the number and diameter of dominant follicles, and the RI and PI values of the subendometrial artery. Conversely, other outcome measures exhibited low to moderate levels of heterogeneity.

### 4.6. Quality Assessment

The quality assessment results are presented in the Summary of Findings Table (Appendix A).

### 4.7. Publication Bias

No evidence of publication bias was detected (Appendix A).

The funnel plots do not suggest publication bias, and Egger’s test result was not significant, indicating that publication bias could not be detected. Significant heterogeneity was observed for ET, the number and diameter of dominant follicles, and the RI and PI values of the subendometrial artery. Other outcome measures exhibited low or moderate heterogeneity.

### 4.8. Risk of Bias Assessment

The results of the risk of bias assessment are displayed in Appendix A.

### 4.9. Strengths and Limitations

We exclusively utilized randomized trials for our meta-analysis. One unique aspect of this meta-analysis is our systematic separation of the results from previous studies involving women with PCOS treated with LE or CC, based on whether patients were monitored for one or more cycles and whether they received a fixed or escalating drug dosage for OI. By doing so, unlike previous meta-analyses, we have minimized the potential for distortion that may arise from quantitative analysis following the amalgamation of these diverse research findings.

For our quantitative analysis, we only included studies that employed fixed drug doses over a single cycle, as these provided sufficient data.

Only a moderate number of studies could be included because the results of patients examined over one cycle could not be aggregated with the total results of patients examined over several cycles, nor with the results of patients whose medication dose was escalated during the examination, if necessary, to ensure accuracy of the data. Furthermore, not all eligible studies employed the same drug dosage of LE or CC.

Another limitation is the presence of moderate and high risk of bias in certain domains.

### 4.10. Implications for Practice and Research

Based on our very encouraging results, we recommend further prospective, higher-quality RCTs on ovulation rate, pregnancy rate, ET, and subendometrial arterial circulation in women with PCOS during LE treatment for OI. These trials will confirm our findings, potentially leading to the introduction of LE into clinical practice as a first-line oral OI drug for patients with PCOS.

In the future, to provide more accurate data, it would be essential to conduct as many research studies as possible with consistent numbers of cycles and accurately record the observed effects with each dosage of LE and CC, especially if the dosage is adjusted during the study.

## 5. Conclusions

Our results indicate that LE is an effective therapeutic option for inducing ovulation in patients with PCOS. When comparing LE with CC, the gold standard drug for OI, we observed higher ovulation rates, thicker endometrium, lower RI of subendometrial arteries, and higher pregnancy rates with the administration of LE. The lower RI in the subendometrial arteries may increase the chance of embryo implantation and improve embryo development. Based on these findings, we recommend further high-quality RCTs to confirm that LE should be the first-line drug for OI in PCOS patients as soon as possible.

## Figures and Tables

**Figure 1 pharmaceuticals-17-00971-f001:**
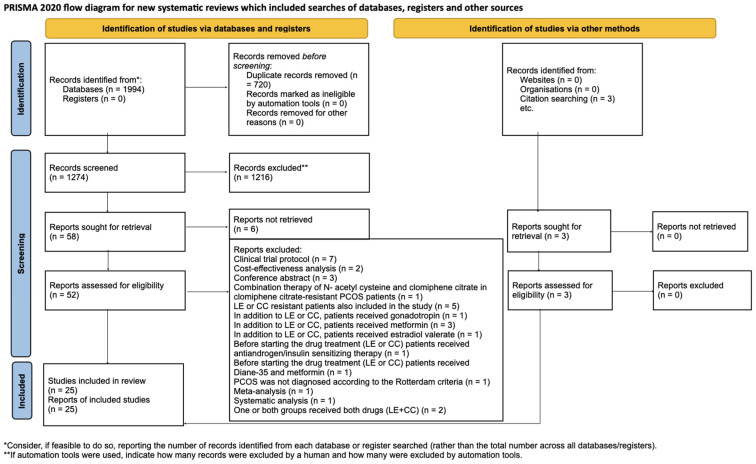
PRISMA [38] flow diagram. The specifics of the search and selection process are presented in detail using a PRISMA [38] flow diagram.

**Figure 2 pharmaceuticals-17-00971-f002:**
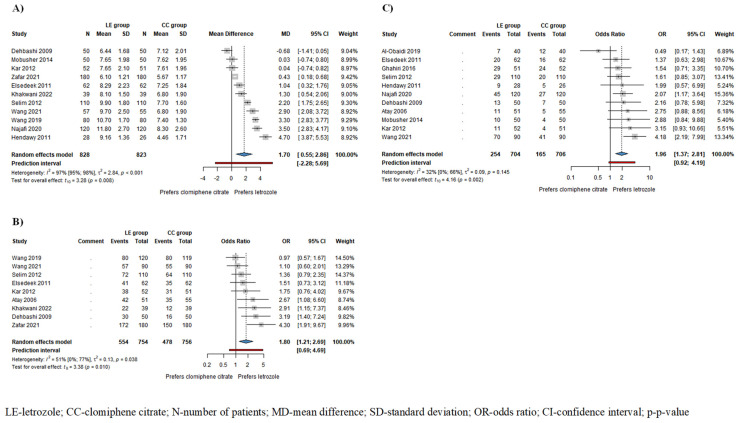
Forest plots for the following outcomes: (**A**) endometrial thickness (ET) in all patients; (**B**) ovulation rate; (**C**) pregnancy rate in all patients. ET, ovulation rate, and pregnancy rate were significantly higher in the LE group compared to the CC group [1,4,5,6,15,16,19,21,22,24,25,26,27,37].

**Figure 3 pharmaceuticals-17-00971-f003:**
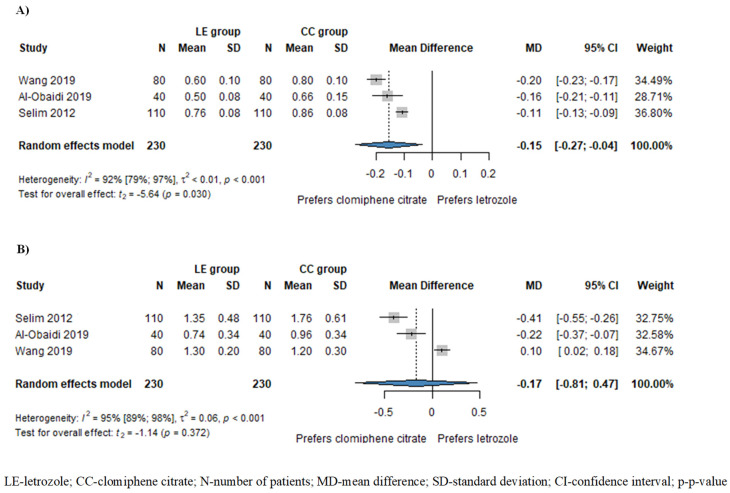
Forest plots for subendometrial circulation: (**A**) resistance index (RI) of subendometrial arteries; (**B**) pulsatility index of subendometrial arteries. The RI of subendometrial arteries was significantly lower in the LE group compared to the CC group. The PI of subendometrial arteries was also lower in the LE group, although this difference did not reach statistical significance [1,15,19].

**Table 1 pharmaceuticals-17-00971-t001:** (**A**): Basic characteristics of the studies included in the systematic review and meta-analysis are presented. (**B**): Basic characteristics of the studies included in the systematic review and meta-analysis are presented.

(**A**)													
**Author**	**Study Site**	**N° of PP Analyzed Patients**	**N° of PP Analyzed Patients in LE Group**		**N° of PP Analyzed Patients in CC Group**	**Follow-up Period (Number of Treatment Cycles)**		**Age (Years)** **in LE Group**	**Age (Years)** **in CC Group**		**BMI** **(kg/m^2^)** **in LE Group**		**BMI** **(kg/m^2^)** **in CC Group**
**Studies included in the meta-analysis**													
Al-Obaidi et al. [15]	Iraq	80	40		40	1		28.45 ± 5.95 ^a^ (PP)	29.58 ± 5.81 ^a^ (PP)		25.29 ± 2.76 ^a^ (PP)		24.87 ± 2.85 ^a^ (PP)
Atay et al. [21]	Turkey	106	51		55	1		27.1 ± 0.9 ^a^ (PP)	26.2 ± 1.1 ^a^ (PP)		26.1 ± 1.91 ^a^ (PP)		25.8 ± 1.77 ^a^ (PP)
Dehbashi et al. [37]	Iran	100	50		50	1		23.62 ± 2.92 ^a^ (PP)	24.32 ± 3.43 ^a^ (PP)		27.45 ± 4.61 ^a^ (PP)		27.09 ± 3.61 ^a^ (PP)
Elsedeek et al. [4]	Egypt	116	59		57	1		24.95 ± 3.11 ^a^ (PP)	25 ± 3.59 ^a^ (PP)		27.7 ± 3.48 ^a^ (PP)		29.18 ± 3.47 ^a^ (PP)
Ghahiri et al. [6]	Iran	101	50		51	1		25.63 ± 4.41 (LE + CC group) ^a^ (PP)	25.63 ± 4.41 (LE + CC group) ^a^ (PP)		28.24 ± 5.2 ^a^ (PP)		27.13 ± 4.9 ^a^ (PP)
Hendawy et al. [22]	Egypt	54	28		26	1		27.2 ± 5.18 ^a^ (ITT)	25.21 ± 5.18 ^a^ (ITT)		26.2 ± 1.8 ^a^ (ITT)		29.1 ± 2.3 ^a^ (ITT)
Hussein et al. [23] ^g^	Iraq	80	40		40	1		28.45 ± 5.95 ^a^ (PP)	29.58 ± 5.81 ^a^ (PP)		25.29 ± 2.76 ^a^ (PP)		24.87 ± 2.85 ^a^ (PP)
Kar [24]	India	103	52		51	1		26.26 ± 2.41 ^a^ (PP)	26.27 ± 2.47 ^a^ (PP)		25.91 ± 3.57^a^ (PP)		25.95 ± 3.31^a^ (PP)
Khakhwani et al. [25]	Pakistan	70	36		34	1		≤20: 5 ^e^ (12.8 ^f^); 21–30: 34 ^e^ (87.2 ^f^) (ITT)	≤20: 3 ^e^ (7.7 ^f^); 21–30: 36 ^e^ (92.3 ^f^) (ITT)		<25: 24 ^e^ (61.5 ^f^); > 25: 15 ^e^ (38.5 ^f^) (ITT)		<25: 28 ^e^ (71.8 ^f^); >25: 11 ^e^ (28.2 ^f^) (ITT)
Morbusher [26]	Pakistan	100	50		50	1		24.29 ± 2.3 ^a^ (PP)	24.26 ± 2.33 ^a^ (PP)		25.91 ± 3.32 ^a^ (PP)		25.89 ± 3.31 ^a^ (PP)
Najafi et al. [16]	Iran	220	110		110	1		26.2 ± 3.6 ^a^ (PP)	27 ± 3.6 ^a^ (PP)		27.6 ± 1.8 ^a^ (PP)		27.3 ± 1.8 ^a^ (PP)
Selim et al. [19]	Saudi Arabia	201	102		99	1		26 ± 2.7 ^a^ (PP)	25.1 ± 3.1 ^a^ (PP)		24.4 ± 4.3 ^a^ (PP)		23.8 ± 3.7 ^a^ (PP)
Wang et al. [1]	China	160	80		80	1		29.2 ± 5.1 ^a^ (PP)	28.4 ± 4.6 ^a^ (PP)		21.4 ± 3.9 ^a^ (PP)		22 ± 3.8 ^a^ (PP)
Wang et al. [5]	China	112	57		55	1		28.5 ± 7.6 ^a^ (ITT)	28.3 ± 7.5 ^a^ (ITT)		24.9 ± 8.4 ^a^ (ITT)		25.3 ± 7.9 ^a^ (ITT)
Zafar et al. [27]	Pakistan	360	180		180	1		26.61 ± 4.81 ^a^ (PP)	27.89 ± 4.24 ^a^ (PP)		NA		NA
**Studies included exclusively in the systematic review**													
						LE group:	CC group:						
Al-Shaikh et al. [28]	Iraq	85	40		45	47	80	18–40 ^d^ (LE + CC group) (PP)	18–40 ^d^ (LE + CC group) (PP)		NA		NA
Amer et al. [29]	The United Kingdom	149	75		74	1 or 6	1 or 6	28.3 ± 4.4 ^a^ (ITT)	28.1 ± 4.2 ^a^ (ITT)		27.5 (23.4–32.2) ^c^ (ITT)		27.7 (23.0–31.0) ^c^ (ITT)
Bansal et al. [36]	India	80	41		39	1 or 3	1 or 3	27.0 ± 3.56 ^a^ (ITT)	26.0 ± 3.97 ^a^ (ITT)		23.90 ± 3.57 ^a^ (ITT)		23.10 ± 3.64 ^a^ (ITT)
Baruah et al. [38]	India	50	25		25	58	56	29.7 ± 0.5 ^a^ (PP)	30.2 ± 0.5 ^a^ (PP)		23.6 ± 0.04 ^a^ (PP)		24.52 ± 0.02 ^a^ (PP)
Bayar et al. [30]	Turkey	74	38		36	99	95	32.2 ±3.9 ^a^ (PP)	30.6± 4 ^a^ (PP)		NA		NA
Legro et al. [31]	The United States	750	374		376	5	5	28.9 ± 4.5 ^a^ (PP)	28.8 ± 4.0 ^a^ (PP)		35.2 ± 9.5 ^a^ (PP)		35.1 ± 9.0 ^a^ (PP)
Ray et al. [35]	India	147	69		78	132	156	28 (19–35) ^c^ (PP)	29 (20–35) ^c^ (PP)		28.8 (23.2–34.6) ^c^ (PP)		28.5 (24.2–33.6) ^c^ (PP)
Roy et al. [32]	India	204	98		106	294	318	26.1 ± 1.8 ^a^ (PP)	26.5 ± 1.3 ^a^ (PP)		25.8 ± 2.1 ^a^ (PP)		25.4 ± 1.56 ^a^ (PP)
Sakar et al. [33]	Turkey	323	175		148	1 or 6	1 or 6	25.9 ± 4 ^a^ (PP)	24.6 ± 4.4 ^a^ (PP)		25.4 ± 3.2 ^a^ (PP)		24.8 ± 2.9 ^a^ (PP)
Sharief et al. [34]	Iraq	75	35		40	6	6	26.1 ± 1.3 ^a^ (PP)	25.3 ± 2.1 ^a^ (PP)		28.1 ± 1.91 ^a^ (PP)		27.8 ± 1.7 ^a^ (PP)
(**B**)													
**Author**	**Duration of Infertility** **(Years) in LE Group**			**Duration of Infertility** **(Years) in CC Group**	**Dosage of LE** **(mg/die)**	**Dosage of CC** **(mg/die)**		**Administration Period of LE (Day of Cycle)**		**Administration Period of LE (Day of Cycle)**		**Trigger of Ovulation with HCG**	**Support of Luteal Phase with Progesterone**
**Studies included in the meta-analysis**													
Al-Obaidi et al. [15]	3.53 ± 1.87 ^a^ (PP)			3.5 ± 1.88 ^a^ (PP)	5	100		3–7		3–7		Y	NA
Atay et al. [21]	2.2 ± 0.7 ^a^ (PP)			2.4 ± 0.9 ^a^ (PP)	2.5	100		3–7		3–7		Y	NA
Dehbashi et al. [37]	2 ± 1.34 ^a^ (PP)			2.3 ± 1.85 ^a^ (PP)	5	100		3–7		3–7		Y	NA
Elsedeek et al. [4]	NA			NA	5	100		3–7		3–7		N	NA
Ghahiri et al. [6]	1: 6 ^e^ (12 ^f^); >1: 44 ^e^ (88 ^f^) (PP)			1: 4 ^e^ (8 ^f^); >1: 47 ^e^ (92 ^f^) (PP)	5	100		3–7		3–7		NA	NA
Hendawy et al. [22]	NA			NA	2.5	100		3–7		3–7		Y	Y
Hussein et al. [23] ^g^	3.53 ± 1.87 ^a^ (PP)			3.5 ± 1.88 ^a^ (PP)	5	100		3–7		3–7		Y	NA
Kar [24]	3.08 ± 1.92 ^a^ (PP)			3.14 ± 2.16 ^a^ (PP)	5	100		2–6		2–6		Y	Y
Khakhwani et al. [25]	<3: 30 ^e^ (76.9 ^f^); ≥3: 9 ^e^ (23.1 ^f^) (ITT)			<3: 30 ^e^ (76.9 ^f^); ≥3: 12 ^e^ (30.8 ^f^) (ITT)	5	100		3–7		3–7		NA	NA
Morbusher [26]	3.18 ± 2.12 ^a^ (PP)			3.12 ± 2.02 ^a^ (PP)	2.5	100		2–6		2–6		Y	Y
Najafi et al. [16]	2.1 ± 1.2 ^a^ (PP)			2.4 ± 1.3 ^a^ (PP)	10	100		3–7		3–7		Y	NA
Selim et al. [19]	2.9 ± 0.6 ^a^ (PP)			2.6 ± 0.7 ^a^ (PP)	5	100		3–7		3–7		Y	NA
Wang et al. [1]	2.4 ± 0.89 ^a^ (PP)			2.1 ± 0.8 ^a^ (PP)	2.5	50		3–7		3–7		Y	NA
Wang et al. [5]	2.4 ± 0.7 ^a^ (ITT)			2.3 ± 0.6 ^a^ (ITT)	2.5	50		5–9		5–9		Y	NA
Zafar et al. [27]	4.11 ± 3.5 ^a^ (PP)			4.7 ± 3.4 ^a^ (PP)	2.5	50		2–6		2–6		NA	NA
**Studies included exclusively in the systematic review**													
Al-Shaikh et al. [28]	NA			NA	5	100		2–5		2–5		NA	NA
Amer et al. [29]	1.5 (1.0–2.0) ^c^ (ITT)			1.5 (1.0–2.0) ^c^ (ITT)	2.5–>5	50–>100		2/4–6/8		2/4–6/8		NA	NA
Bansal et al. [36]	3.9 ± 2.3 ^a^ (ITT)			3.4 ± 2.3 ^a^ (ITT)	2.5–>7.5	50–>150		2–6		2–6		Y	NA
Baruah et al. [38]	2.7 ± 0.2 ^a^ (PP)			2.9 ± 0.5 ^a^ (PP)	2.5–>5	50–>100		5–9		5–9		Y	NA
Bayar et al. [30]	5 (1–10) ^c^ (PP)			3 (1–11) ^c^ (PP)	2.5	100		3–7		3–7		Y	NA
Legro et al. [31]	40.9 ± 38.0 ^a^ (PP)			42.5 ± 37.6 ^a^ (PP)	2.5–>7.5	50–>150		3–7		3–7		NA	NA
Ray et al. [35]	2.2 ^b^ (PP)			2.4 ^b^ (PP)	2.5	100		3–7		3–7		Y	NA
Roy et al. [32]	6.4 ± 3.8 ^a^ (PP)			5.8 ± 3.1 ^a^ (PP)	2.5–>5	50–>100		3–7		3–7		Y	NA
Sakar et al. [33]	2 (1–12) ^c^ (PP)			2 (1–11) ^c^ (PP)	5	100		2–5		2–5		NA	NA
Sharief et al. [34]	2.4 ± 0.6 ^a^ (PP)			2.3 ± 0.4 ^a^ (PP)	2.5–5	100–200		3–7		3–7		Y	NA

^a^: parameters are presented as mean with standard deviation; ^b^: parameters are presented as mean; ^c^: parameters are presented as median (min-max); ^d^: parameters are presented as min-max; ^e^: number of patients; ^f^: % of patients. ^g^: The patient population under investigation aligns with that of Al-Obaidi et al. [15], with individuals referenced in the article. Consequently, we have incorporated these findings solely once within our meta-analysis; ITT: the data were analyzed according to the intention-to-treat principle; PP: the data were analyzed on a per-protocol basis; N°: number; LE: letrozole; CC: clomiphene citrate; BMI: body mass index; NA: no data available; Y: yes; N: no.

**Table 2 pharmaceuticals-17-00971-t002:** (**A**): Conclusions drawn from the studies exclusively included in the systematic review. (**B**)**:** Conclusions drawn from the studies exclusively included in the systematic review.

(**A**)										
**Author**	**Endometrial Thickness**		**Endometrial Volume**	**Multilayered Endometrial Pattern and/or Echogenicity**	**N° of Dominant Follicles**	**Diameter of Dominant Follicles**	**Monofollicular Development Cycles**	**Multifollicular Development Cycles**	**Ovulation Rate**	**Pregnancy Rate**
Al-Obaidi et al. [15]	significantly higher in the LE group ^l^		NA	NA	j	j	NA	NA	NA	j
Al-Shaikh et al. [28]	significantly higher in the CC group		NA	NA	significantly higher in the LE group ^h^	no significant difference between the LE and CC groups ^h^	NA	NA	NA	no significant difference between the LE and CC groups ^h^
Amer et al. [29]	significantly higher in the CC group		NA	no significant difference between the LE and CC groups	NA	NA	NA	NA	no significant difference between the LE and CC groups; however, upon analyzing the data per cycle, a significantly higher outcome was observed in the LE group	significantly higher in the LE group
Bansal et al. [36]	no significant difference between the LE and CC groups		NA	NA	NA	NA	significantly higher in the LE group	NA	no significant difference between the LE and CC groups ^i^	significantly higher in the LE group
Baruah et al. [38]	significantly higher in the LE group		NA	NA	no significant difference between the LE and CC groups	NA	NA	NA	NA	no significant difference between the LE and CC groups ^h^
Bayar et al. [30]	similar between the LE and CC groups		NA	NA	significantly lower in the LE group	NA	NA	NA	NA	no significant difference between the LE and CC groups ^h^
Ghahiri et al. [6]	NA		NA	NA	NA	NA	NA	NA	NA	j
Hussein et al. [23]	significantly higher in the LE group ^m^		NA	the occurrence of a multilayered endometrial pattern and hypoechogenic endometrium was significantly higher in the letrozole (LE) group; moreover, hypoechogenic endometrium was highly associated with pregnancy in both groups	k	k	NA	NA	NA	k
Legro et al. [31]	significantly higher in the CC group		NA	NA	NA	NA	NA	NA	significantly higher in the LE group	significantly higher in the LE group
Ray et al. [35]	significantly higher in the LE group		NA	NA	no significant difference between the LE and CC groups ^h^	no significant difference between the LE and CC groups	majority of induced cycles in both the LE and CC group	NA	NA	no significant difference between the LE and CC groups
Roy et al. [32]	significantly higher in the LE group		NA	NA	no significant difference between the LE and CC groups	NA	NA	NA	similar between the LE and CC groups	no significant difference between the LE and CC groups ^h^
Sakar et al. [33]	significantly higher in the LE group		NA	NA	similar between the LE and CC groups	NA	NA	NA	significantly higher in the LE group ^h^	significantly higher in the LE group
Sharief et al. [34]	significantly higher in the LE group		NA	NA	significantly lower in the LE group	NA	NA	NA	significantly higher in the LE group	no significant difference between the LE and CC groups
Wang et al. [1]	j		significantly higher in the LE group	the occurrence of a multilayered endometrial pattern was significantly more frequent in the LE group	j	j	NA	NA	j	^l^
Wang et al. [5]	j		significantly higher in the LE group	NA	j	j	NA	NA	j	j
Dehbashi et al. [37]	j		NA	NA	j	NA	NA	NA	j	j
Selim et al. [19]	j		NA	NA	j	NA	NA	NA	j	j
(**B**)										
**Author**	**N° of Multiple Pregnancies**	**N° of Miscarriages**	**Live Birth Rate**	**Prevalence of Ectopic Pregnancy**	**N° of Fetal Anomalies**	**Endometrial VI, FI, VFI, or Detection Rate of Endometrial-Subendometrial Blood Flow**	**RI and PI of Subendometrial Arteries**	**SV/DV of Subendometrial Arteries**	**RI and PI of Uterine Arteries**	**VEGF and/or Integrin alpha vß3 Concentration in Uterine Fluid**
Al-Obaidi et al. [15]	NA	NA	NA	NA	NA	NA	NA	significantly higher in the LE group	NA	VEGF exhibited a significantly negative correlation with endometrial RI in the letrozole (LE) group, while no such correlation was observed in the clomiphene citrate (CC) group
Al-Shaikh et al. [28]	NA	one patient in the LE group and two in the CC group	NA	NA	NA	NA	NA	NA	NA	NA
Amer et al. [29]	NA	NA	no significant difference between the LE and CC groups	NA	none in the LE group and none in the CC group	NA	NA	NA	NA	NA
Bansal et al. [36]	NA	NA	NA	NA	NA	NA	NA	NA	NA	NA
Baruah et al. [38]	none in the LE group and nine in the CC group	NA	NA	NA	NA	NA	significantly lower in the LE group	NA	NA	NA
Bayar et al. [30]	none in the LE and none in the CC groups	one patient in the LE group	no significant difference between the LE and CC groups ^h^	NA	NA	NA	NA	NA	NA	NA
Ghahiri et al. [6]	j	j	NA	no significant difference between the LE and CC groups	NA	NA	NA	NA	NA	NA
Hussein et al. [23]	NA	NA	NA	NA	NA	NA	NA	NA	NA	NA
Legro et al. [31]	no significant difference between the two groups	no significant difference between the LE and CC groups	significantly higher in the LE group	NA	four patients in the LE group and one in the CC group	NA	NA	NA	NA	NA
Ray et al. [35]	NA	one patient in the CC group	NA	NA	none in the LE group and none in the CC group	NA	NA	NA	NA	NA
Roy et al. [32]	none in the LE group and three patients in the CC group	no significant difference between the LE and CC groups	significantly higher in the LE group	NA	NA	NA	NA	NA	NA	NA
Sakar et al. [33]	none in the LE group and none in the CC group	no significant difference between the LE and CC groups	significantly higher in the LE group	NA	one patient in the LE group and one in the CC group	NA	NA	NA	NA	NA
Sharief et al. [34]	NA	NA	NA	NA	NA	NA	NA	NA	NA	NA
Wang et al. [1]	NA	NA	NA	NA	NA	endometrial VI, FI, and VFI were significantly higher in the LE group	j	NA	no significant difference between the LE and CC groups	NA
Wang et al. [5]	NA	NA	NA	NA	NA	endometrial VI, FI, and VFI were significantly higher in the LE group	NA	NA	no significant difference between the LE and CC groups	significantly higher in the LE group
Dehbashi et al. [37]	j	j	no significant difference between the LE and CC groups	NA	none in the LE group and one in the CC group	NA	NA	NA	NA	NA
Selim et al. [19]	j	NA	NA	NA	NA	the rate of endometrial–subendometrial blood flow was significantly lower in the letrozole (LE) group	j	NA	NA	NA

N°: number; LE: letrozole; CC: clomiphene citrate; ^h^: the data are presented per cycle; ^i^: cumulative ovulation rate; j: the data are included in the quantitative analysis in the meta-analysis (refer to the Section 2); k: the data are included in the quantitative analysis in the meta-analysis (the studied patient population is identical to that in the article by Al-Obaidi et al., 2019 [15]; therefore, we only included these results once in our meta-analysis. Please refer to the results of Al-Obaidi et al. [15]); ^l^: the provided results could not be utilized for our quantitative analysis; thus, we did not incorporate them into our meta-analysis; ^m^: the patient population studied is identical to that in the article by Al-Obaidi et al., 2019 [15]. However, the provided results could not be utilized for our quantitative analysis. Consequently, we did not incorporate them into our meta-analysis. Instead, we included these results in our qualitative analysis within the systematic review. VI: vascularization index; FI: flow index; VFI: vascularization flow index; RI: resistance index; PI: pulsatility index; SV: systolic velocity; DV: diastolic velocity; VEGF: vascular endothelial growth factor.

## Data Availability

Data are contained within the article or Appendix A.

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
