# Peer review of "Strong Early Impact of Letrozole on Ovulation Induction Outperforms Clomiphene Citrate in Polycystic Ovary Syndrome"

_pharmaceuticals, 2024, doi:10.3390/ph17070971_

Round 1

Reviewer 1 Report

Comments and Suggestions for Authors

The authors reviewed and analysed nearly 30 studies to investigate the effects of letrozole and clomiphene citrate on PCOS. The difference in efficacy between two medicines was shown. The following points should be addressed or improved.

*Table 1 & 2: Tables should be reformatted. Too difficult to read.

*Figure 1: too low resolution for reading

*Selection of paper. Although the authors claimed no biases were detected, nearly all the studies were conducted in middle east countries and China. Please explain why studies from other sites did almost not exist.

Author Response

Comment 1: Table 1 & 2: Tables should be reformatted. Too difficult to read.

Response 1:  Thank you for pointing this out. I have changed tables 1 (page numbers: 4-7) and 2 (pane numbers: 11-17) as it was requested.

Comment 2: Figure 1: too low resolution for reading

Response 2:  Thank you for pointing this out.  As you requested, I replaced the figures in the manuscript with better-quality ones (page number: 3; line: 88 and page number: 8; line: 135; and page number: 9; line: 147).

Comment 3: Selection of paper. Although the authors claimed no biases were detected, nearly all the studies were conducted in middle east countries and China. Please explain why studies from other sites did almost not exist.

Response 3: We did not deal with which populations the articles collected data from during the selection process.The articles that meet the strict requirements we examined are included in the systematic review and meta-analysis we prepared. Most of them originated from China and the Middle East. In the "limitations" section, we also explain that unfortunately only a moderate number of studies could be included. This is because the results of the patients examined over one cycle could not be aggregated with those examined over several cycles, nor with the results of patients whose drug dose was increased during the study to ensure data accuracy.

Reviewer 2 Report

Comments and Suggestions for Authors

Manuscript ID: pharmaceuticals-3063741

Title “Strong early impact of letrozole on ovulation induction outperforms clomiphene citrate in polycystic ovary syndrome

Journal: Pharmaceuticals

Vajna et al. reported a systematic review and meta-analysis to explore the efficacy of letrozole compared with clomiphene citrate for ovulation induction in women with PCOS. The study is important as for as the efficacy and health risks of letrozole are concerned in women with Polycystic ovary syndrome; however there are some issues in the study which need to be rectified before publication.  

1.      The rationale and novelties of the study must be properly described in the introduction section as plenty of work is reported on the topic. Discuss your work in the context of the reported work on the topic.

2.      The first para of section 2.3 on page 9, needs to be rephrased to make it clear?

3.      Figure 1 has not been properly placed in the text while others have been placed. The quality (resolution) of the figures needs to be increased. Table 1 is so lengthy and confusing. Make it simple and concise?    

4.      Mention the complete name on first mention such as RI, and PI. Avoid using the abbreviations in the abstract. 

5.      For how much time the two treatments were followed? It should be clarified in the methodology sections of the manuscript?

6.      To know about the real picture of the study the drug dosage and duration of study of LE or CC in all the studies may be same?. 

7.      The correlation must be established between the various outcome parameters of two treatments in result section through graphical representation.

8.      The methodology section needs to be more descriptive to reproduce this work easily.

9.      The discussion section must be properly summarized and the outcomes of the study must be organized properly to make it simple?.

10.  What suggestions and strategies will be recommended to the health care providers in decision-making and holding effective interventions with letrozole?   

11.  The manuscript must be revised as per journal approved format. The up to date references must be incorporated and grammatical and typographical mistakes must be rectified.  Manuscript ID: pharmaceuticals-3063741

Title “Strong early impact of letrozole on ovulation induction outperforms clomiphene citrate in polycystic ovary syndrome

Journal: Pharmaceuticals

Vajna et al. reported a systematic review and meta-analysis to explore the efficacy of letrozole compared with clomiphene citrate for ovulation induction in women with PCOS. The study is important as for as the efficacy and health risks of letrozole are concerned in women with Polycystic ovary syndrome; however there are some issues in the study which need to be rectified before publication.  

1.      The rationale and novelties of the study must be properly described in the introduction section as plenty of work is reported on the topic. Discuss your work in the context of the reported work on the topic.

2.      The first para of section 2.3 on page 9, needs to be rephrased to make it clear?

3.      Figure 1 has not been properly placed in the text while others have been placed. The quality (resolution) of the figures needs to be increased. Table 1 is so lengthy and confusing. Make it simple and concise?    

4.      Mention the complete name on first mention such as RI, and PI. Avoid using the abbreviations in the abstract. 

5.      For how much time the two treatments were followed? It should be clarified in the methodology sections of the manuscript?

6.      To know about the real picture of the study the drug dosage and duration of study of LE or CC in all the studies may be same?

7.      The correlation must be established between the various outcome parameters of two treatments in result section through graphical representation.

8.      The methodology section needs to be more descriptive to reproduce this work easily.

9.      The discussion section must be properly summarized and the outcomes of the study must be organized properly to make it simple?.

10.  What suggestions and strategies will be recommended to the health care providers in decision-making and holding effective interventions with letrozole?   

11.  The manuscript must be revised as per journal approved format. The up to date references must be incorporated and grammatical and typographical mistakes must be rectified.  

Comments on the Quality of English Language

Minor changes recommended

Author Response

Comment 1: The rationale and novelties of the study must be properly described in the introduction section as plenty of work is reported on the topic. Discuss your work in the context of the reported work on the topic.

Response 1: Thank you for pointing this out. In point 4.9 of our article (page number: 23; line: 419), the "strengths" section, we discuss in detail why the study we prepared is unique and of a higher quality than previous studies on the same topic. Compared to previously published studies, a fundamental innovation in our article is that we specifically examined the early effect of letrozole versus clomiphene citrate. To get a more accurate picture of the mechanism of action of the two active substances, we only compared studies in which only clomiphene citrate or letrozole was administered to patients with PCOS and in which the drugs mentioned above were used in a fixed dose during one cycle. Previous studies published on a similar topic showed a significant deviation in terms of drug dose and therapy, which does not allow for an exact comparison of these active substances and distorts the quantitative analysis.

Comment 2: The first para of section 2.3 on page 9, needs to be rephrased to make it clear?

Response 2: Thank you for pointing this out. I have reworded the first paragraph of section 2.3. (page number: 8; lines: 118-124).

Comment 3: Figure 1 has not been properly placed in the text while others have been placed. The quality (resolution) of the figures needs to be increased. Table 1 is so lengthy and confusing. Make it simple and concise?    

Response 3: Thank you for pointing this out.  I have moved Figure 1 (page number: 3; line: 86) to the appropriate place in the text, increased the quality of the figures (page number: 8; line: 133 and page number: 9; line: 145), and converted Table 1 (page numbers: 4-7).

Comment 4: Mention the complete name on first mention such as RI, and PI. Avoid using the abbreviations in the abstract. 

Response 4: Thank you for pointing this out. I have avoided abbreviations in the abstract, as you requested (page number: 1; lines: 21-28).

Comment 5: For how much time the two treatments were followed? It should be clarified in the methodology sections of the manuscript?

Response 5: Patients suffering from PCOS were followed for one cycle. With the help of these articles, we had the opportunity to examine the early effect of letrozole (I have indicated this in Table 1A /page number: 4/ and also indicated it in the "methods" section according to your request /page number: 20; lines: 324-326/).

Comment 6: To know about the real picture of the study the drug dosage and duration of study of LE or CC in all the studies may be same?

Response 6: Table 1A (page nuber: 4) shows the drug doses used at the end of the study per study (in each article, patients with PCOS were studied for one cycle), and Table 1B (page number: 6) shows symptoms per study (a fixed drug dose was used in all studies included in our quantitative analysis). In clinical practice, fixed doses are usually used for stimulation. Dosing determined per kg of body weight was not widespread. Therefore, we think that the simultaneous comparison of the use of letrozole and clomiphene citrate at different doses in different studies may give an inaccurate picture of effectiveness. Based on this, we decided to use those studies that used the fixed same drug doses over one cycle for our quantitative analysis.

Comment 7: The correlation must be established between the various outcome parameters of two treatments in result section through graphical representation.

Response 7: The most important results are implemented in the text (page nuber: 8; line 141 and page number: 9; line: 153), and the supplementary material provides a graphical representation of all results, as requested.

Comment 8: The methodology section needs to be more descriptive to reproduce this work easily.

Response 8: Thank you for pointing this out. As requested, we added more details to the "methods" section (page numbers: 20-23; lines: 314-426).

Comment 9: The discussion section must be properly summarized and the outcomes of the study must be organized properly to make it simple?

Response 9: Thank you for pointing this out. Per your request, we have summarized and simplified the "discussion" section (page numbers: 18-20; lines: 208-312).

Comment 10: What suggestions and strategies will be recommended to the health care providers in decision-making and holding effective interventions with letrozole?  

Response 10: Based on our research, we have observed that letrozole’s ovulation induction effect is promising, and even in its early stages, it seems to be more effective than clomiphene citrate. However, to raise this recognition to the level of evidence and treat letrozole as a first-line ovulation induction agent, additional high-quality randomized controlled trials are needed to confirm our results.

Comment 11: The manuscript must be revised as per journal approved format. The up to date references must be incorporated and grammatical and typographical mistakes must be rectified.  Manuscript ID: pharmaceuticals-3063741

Response 11: Thank you for pointing this out. I have corrected the manuscript as requested (page numbers: 1-28).

Reviewer 3 Report

Comments and Suggestions for Authors

The systematic review and meta-analysis evaluates the efficacy of letrozole compared to clomiphene citrate for ovulation induction in women with polycystic ovary syndrome (PCOS), finding that letrozole significantly increases endometrial thickness, ovulation rates, and pregnancy rates while also improving subendometrial blood flow. These results suggest that letrozole may be a more effective first-line treatment for ovulation induction in PCOS patients than the currently favored clomiphene citrate. Here are some comments that would strengthen the review:

-In the abstract, the databases used in the search method should be included.

-Introduction, line 42-43, could you please clarify the prevalence of PCOS, the 9%-18% of what?

-In the methodology, the authors did not give the rationale for including just the RCTs.

-In the risk of bias assessment, can you briefly describe the ROB2 tool, which is the criteria used in this tool.

-In the results, the authors should compare the confounding factors such as BMI, age to see if there is an effect on the results.

-The figures are of low quality, please make them more clear.

-Detailed legends for the figures are required.

-Although the authors stated that the results have high heterogeneity, they should discuss this more and clarify the source of this heterogeneity and the effect of it on their conclusion.

Comments on the Quality of English Language

Author Response

Comment 1: In the abstract, the databases used in the search method should be included.

Response 1: Thank you for pointing this out. As you requested, I also included the databases used in the search method in the abstract (page number: 1; lines: 26-28).

Comment 2: Introduction, line 42-43, could you please clarify the prevalence of PCOS, the 9%-18% of what?

Response 2:  Thank you for pointing this out. As you requested, I clarified the prevalence of PCOS in the "introduction" section (page number: 2; line: 45).

Comment 3: In the methodology, the authors did not give the rationale for including just the RCTs.

Response 3: Our working group, which has already conducted numerous meta-analyses, operates based on stringent methodological rules to achieve high quality and reduce distortions as much as possible. Due to the strict criteria, we could only select the already mentioned RCTs after examining all possible articles. Unfortunately, articles that are not prepared according to such strict requirements can significantly impair the quality of the final evidence. To contribute to the potential change of current clinical practice with the results of our research work, a meta-analysis should only be made of high-quality articles.

Comment 4: In the risk of bias assessment, can you briefly describe the ROB2 tool, which is the criteria used in this tool.

Response 4: Thank you for pointing this out. As you requested, I briefly explained the criteria of the ROB2 device in point 4.3 (page number: 21; lines: 357-362).

Comment 5: In the results, the authors should compare the confounding factors such as BMI, age to see if there is an effect on the results.

Response 5. In clinical practice, fixed starting doses have spread, where age, BMI, etc., are not primarily evaluated. Therefore, the data were implemented in the table to give a more accurate picture of each examination, but this was not an exclusion criterion.

Comment 6: The figures are of low quality, please make them more clear.

Response 6: Thank you for pointing this out.  I replaced the figures in the manuscript with better-quality ones per your request (page number: 3; line: 88 and page number: 8; line 135 and page number 9; line: 147).

Comment 7: Detailed legends for the figures are required.

Response 7: Thank you for pointing this out. I wrote a detailed legend for the figure shown in the manuscript, as you requested (page number: 3; lines: 101-102 and page number: 8; lines: 136-138 and page number 9; line: 162-165).

Comment 8: Although the authors stated that the results have high heterogeneity, they should discuss this more and clarify the source of this heterogeneity and the effect of it on their conclusion.

Response 8:  Thank you for pointing this out. Although we tried to compare objective parameters, the subjective judgment of the person performing the ultrasound examination is also essential when measuring the endometrium's thickness and determining the follicles' size. This is not only due to the inter-observer error, but there needs to be a consensus on what is considered a dominant follicle based on objective parameters. Furthermore, few studies could be compared well regarding the resistance index and pulsatility index of the subendometrial arteries. When examining the circulation of the subendometrial arteries, the possibility of intra- and inter-observer errors is also magnified. Nevertheless, our conclusions are very forward-looking and raise interesting questions, but also shed light on the question of the accuracy of the tests. It is challenging to eliminate the subjective factor (the subjective judgment of the examiner). In the future, international consensus and coordination at the accepted level will be necessary for conducting examinations. Thank you for bringing this to our attention.

Round 2

Reviewer 3 Report

Comments and Suggestions for Authors

NONE